DATA RELEASE

# AI in practice: a multilingual survey of 2025 BioHackathon participants

Nattawet Sriwichai[1,†], Lucas Feriau[2,†], Pumipat Tongyoo[3,4], Yukiko Noda[2], Hikaru Gyoji[2], Pitiporn Noisagul[1], Susumu Goto[5], David Steinberg[6,*] and Chatarin Wangsanuwat[7,*]

1 Center of Multidisciplinary Technology for Advanced Medicine (CMUTEAM), Faculty of Medicine, Chiang Mai University, Thailand
2 PENQE, Inc., Japan
3 Center for Agricultural Biotechnology, Kasetsart University, Kamphaeng Saen Campus, Thailand
4 Center of Excellence on Agricultural Biotechnology: (AG-BIO/MHESI), Kasetsart University, Thailand
5 Research Organization of Information and Systems, Joint-Support Center for Data Science Research, Database Center for Life Science, Japan
6 Camber, San Francisco, USA
7 Siriraj Long-read Lab (Si-LoL), Medical Bioinformatics, Siriraj Genomics, Faculty of Medicine Siriraj Hospital, Mahidol University, Thailand

## ABSTRACT

This dataset arises from a multilingual survey of AI use among participants and community members in the DBCLS BioHackathon 2025 in Japan. The questionnaire, offered in English, Japanese, and Thai, asked about how often respondents use AI tools, what they use them for, obstacles they encounter, institutional support, satisfaction, and concerns. Additional items captured role, institution type, work country, and other demographics, totaling 105 responses. The dataset includes both raw anonymized responses and a cleaned, standardized English-only version suitable for quantitative analysis, along with the full questionnaire, a data dictionary for cleaned dataset, and a translation lookup table. Free-text answers were screened and redacted to remove URLs, names, and other potentially identifiable information. Together, these materials provide a community-level view of AI practice in genomics, bioinformatics, software development, and related areas, and can support work on AI adoption, policy, and methods for analyzing survey data on AI use in science.

**Subjects** Software and Workflows, Bioinformatics, Machine learning

**Submitted:** 28 November 2025

\* Corresponding authors. E-mail: david@resium.com; chatarin.wan@mahidol.ac.th

† Contributed equally.

Preprint submitted at https://doi.org/10.64898/2026.03.25.713611

## DATA DESCRIPTION

### Context

The survey and initial analyses are reported in a preprint ("AI in practice: insights from a community survey of BioHackathon participants", BioHackrXiv/OSF [1]). This Data Release instead documents the anonymized multilingual dataset and its structure to support reuse [2].

This dataset captures how members of the BioHackathon community and their wider networks are using AI tools in their day-to-day work and, to a lesser extent, daily life. It draws on an online questionnaire offered in English, Japanese, and Thai to participants and community members linked to the 2025 BioHackathon in Japan. It captures how

**Figure 1.** Summary of the BioHackathon 2025 AI Survey Dataset. The dataset captures quantitative metrics of AI adoption and sentiment (*N* = 105 finalized responses). (A) Distribution of overall self-reported AI usage levels. (B) Top participating countries (displaying countries with *n* > 1). (C) Self-reported AI task usage and capacity. Respondents indicated whether they use AI to "Assist" (help with specific parts), "Draft" (generate a first draft for editing), or "Complete" (finish the task with little to no editing). (D) Average concern levels regarding the integration of AI tools, measured on a 1-to-5 scale (1 = "Not concerned at all", 5 = "Very concerned").

respondents use or avoid AI tools in their work, the kinds of support and problems they report, and their concerns about potential harms. Recruitment through participants' networks brought in respondents from a range of countries and institution types, providing a community-level snapshot of "AI in practice" around the BioHackathon community. A high-level summary of this dataset, highlighting overall adoption, geographic reach, task-specific utilization, and AI-related concerns, is present in Figure 1.

## METHODS

### Survey design and implementation

The survey was implemented in Google Forms and designed to take around 10–15 minutes to complete. Respondents first saw an introduction explaining that the goal was to understand how BioHackathon participants use AI in their work and to build a community resource that would be shared back in aggregated form. A working definition of "AI" was provided, covering machine learning, generative AI, scientific AI tools, and AI-powered tools for coding, data analysis, or literature review, while excluding traditional statistical methods unless they involved AI or machine learning.

The survey covered the following domains:

- Demographic information: BioHackathon participation background, interested fields, institution type(s), country of work, age range, and gender.
- Overall AI usage: frequency and style of AI use (e.g. daily user, infrequent user, AI researcher/developer), and reasons for not using AI where applicable.



- Applications: main AI tools used (free-text), and task-specific usage for coding, research, brainstorming, writing/editing, teaching and curriculum, translation, and personal use. For each task, respondents indicated whether they used AI primarily to "assist", "draft", "complete", or not use AI for that task.
- Challenges and satisfaction: challenges encountered when using AI tools, overall satisfaction with AI use, and areas where AI tools would need to improve for respondents to use them more confidently.
- Institutional support and harms: perceived level of institutional support, types of support provided, and concern points (algorithm bias, data privacy/security, intellectual property/ownership, misinformation/hallucinations, and environmental impact).
- Open-ended narratives: several free-text items capturing brief work descriptions, AI success/failure stories, and final comments (included in the raw dataset only).

To reduce pressure and potential bias, only two items were minimally required (survey language and an AI usage-level question). All other items were optional, and respondents could stop participation at any time by closing the form.

The survey supported English, Japanese, and Thai languages. Controlled-response options (e.g. field, institution type, challenges, harms) were translated by the author team members with native fluency, and a translation lookup table was created to align response categories across languages.

## Anonymization and cleaning

The survey was designed to avoid direct identifiers from the outset. It did not request names, institutional affiliations, or IP addresses, and login accounts were not stored in the exported data. To ensure participation privacy while maximizing dataset utility, our processing approach was informed by the Safe Harbor de-identification guidelines outlined under the HIPAA Privacy Rule [3]. One optional item asked for an email address for possible follow-up; no follow-up has been conducted and all email addresses were removed before release. A second optional item invites links to respondents' work (e.g. papers, repositories); this question and all responses to it are excluded from the shared dataset. For the raw anonymized datasets, original responses are preserved as-is, with the exception of the redactions and exclusions described below.

Because seemingly anonymous details can inadvertently identify a person when pieced together [4], free-text responses were reviewed and processed as follows:

- All URLs in free-text responses were redacted as [URL redacted].
- Text that could reasonably identify an individual was removed.

For cleaning, in the cleaned English dataset, the following was applied to produce a robust, structured dataset [5]:

- Long narrative free-text items (e.g. AI success/failure stories, final comments) were dropped to reduce re-identification risk and to provide a compact analysis-ready table, while structured variables and short categorical responses were retained.
- Categorical "Other" responses: For questions allowing "Other" free-text input, the predefined categorical choices were extracted, while the respondent's custom text was preserved in dedicated, adjacent companion columns (e.g., field_other_text) to maintain tabular structure.



- Country standardization: Free-text country entries were standardized to resolve common spelling and abbreviation variations (e.g., mapping "U.K." to "United Kingdom", "germany" to "Germany"). Clearly invalid entries (e.g. "Postdoctoral researcher") were set to missing, while the entry "Global" was retained as-is to reflect the respondent's specific work context.
- AI Tool grouping: Free-text lists of AI tools were parsed using pattern matching to standardize names and correct misspellings (e.g., mapping "chatGTP" or "chat gpt" to "ChatGPT", and "germini" to "Gemini"). Ambiguous product names were generalized (e.g., "copilot" mapped to "Copilot (unspecified)" to account for both Microsoft and GitHub variants), while "VSCode copilot" was grouped to "GitHub Copilot". Generic terms (e.g., "LLMs", "open source"), non-AI tools, and unrelated narrative fragments (e.g., "autocorrect", "Unfortunately a couple …") were filtered out.
- Missing data normalization: Responses of "Prefer not to answer" (e.g., in Age and Gender demographics) were converted to standard missing values (blanks/NAs).

These cleaning and standardization steps were not applied to the raw datasets to ensure the original survey responses remain available for qualitative research or research of other use (Figure 2).

## Data records

The dataset is organized into three main components: raw anonymized responses, a cleaned English-only table, and supporting documentation.

- Raw anonymized data (original languages) "Biohackathon2025AISurvey_data_raw_anon_original_languages_v1.csv": Anonymized responses in the original survey languages (English, Japanese, Thai). This file includes all structured items plus short free-text responses, with URLs redacted and the optional email and citation-link questions removed.
- Raw anonymized data (English) "Biohackathon2025AISurvey_data_raw_anon_ENG_v1.csv": An English translation of the anonymized raw data. Translation was aligned using the translation lookup table.
- Cleaned data (English) "Biohackathon2025AISurvey_data_cleaned_ENG_R1.csv": Cleaned English dataset suitable for quantitative analysis. Long narrative free-text questions are excluded, and categorical "Other" free-text inputs are preserved in adjacent companion columns. Column headers have been shortened and formatted for easy coding via snake case, and responses like "prefer not to answer" were converted to blank cells.
- Codebook (English) "Biohackathon2025AISurvey_cleaned_codebook_ENG_R1.csv": A comprehensive data dictionary for the cleaned English dataset. It provides a 1-to-1 mapping of the 33 variables to their original survey questions. For each variable, the codebook also lists a description, data type, and example value(s).
- Questionnaire "Biohackathon2025AISurvey_GoogleForms.pdf": PDF export of the Google Forms survey, including all sections and skip logic as implemented, with English, Japanese, and Thai questions in their original layout.
- Translation mapping "Translation_lookup_table_R1.csv": A lookup table aligning controlled-response options across English, Japanese, and Thai, and providing translations of short free-text responses from Japanese and Thai into English. It was used to harmonize response categories and support construction of the English-only and cleaned datasets.



## Biohackathon 2025 survey data

N = 105 finalized responses across English, Japanese, and Thai

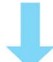

## De-identification & redaction

Removed email/citatio links. Removed identifying narratives.

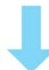

## Raw anonymized datasets

Original free-text retained. No spelling, formatting, or country standardization applied. Preserved for qualitative reuse.

Outputs: data_raw_anon_original_languages & data_raw_anon_ENG

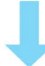

## Analysis-ready datasets

Dropped long narrative texts, standardized country names, grouped AI tools, extracted categorical "other" text

Outputs: data_cleaned_ENG

**Figure 2.** Data processing workflow. All 105 finalized survey responses underwent an initial de-identification to remove potential identifiers and URLs, resulting in the raw anonymized datasets. The raw English data was then subjected to further processing to produce analysis-ready cleaned dataset for quantitative use.

## Data validation and quality control

Several checks were performed to ensure internal consistency and to reduce disclosure risk:

- Response inclusion: The 105 rows in the dataset represent all finalized and submitted responses. Because the survey was administered via Google Forms, which does not track session abandonment or partial progress, the total number of surveys attempted is unavailable.
- Data cleaning and deduplication: All submitted responses were inspected for duplicates and entirely empty submissions. No duplicate entries or entirely blank forms were identified in the exported data; therefore, no responses were excluded from the reported dataset.
- Cross-language consistency: For controlled-response questions, English, Japanese, and Thai options were translated and reviewed by native Japanese and Thai speakers. A translation lookup table from this process was used to ensure that equivalent response options mapped to the same underlying concept across languages.

- Structural consistency: Row counts were verified to match between raw and cleaned English data, and the removal of long free-text columns was checked to confirm that no other variables were inadvertently dropped.
- Survey skip logic: High rates of missing data for certain items are an expected result of the survey design. The question "Is there a reason why you don't use AI?" was only presented to respondents who indicated they do not use AI. To reduce survey fatigue, the follow-up "What would AI tools need to improve?" was only presented to respondents who reported lower overall satisfaction (scores 1–3) and was skipped for satisfied users (scores 4–5).
- Missing data and response rates: To reduce respondent pressure and potential bias, only the survey language and an AI usage-level question were required. All other items were optional. This design resulted in high missing data rates, particularly for domain-specific items (e.g., "Teaching and curriculum") where the topic may have been irrelevant to the respondent's role. Re-users of this dataset should handle missing values appropriately, as they are likely not missing completely at random.
- Task usage limitations: In addition to the optional nature of the items, interpretations of task-specific AI usage are limited by the available response options. The survey provided three active choices ("Assist", "Draft", or "Complete") but lacked explicit "Not used" or "Not applicable" options. Consequently, a missing response conflates three distinct states: the respondent does not use AI for that task, the task is not applicable to their professional role, or the respondent simply chose not to answer.
- AI concern items: The questions regarding concerns about AI (e.g., bias, data privacy) were evaluated on a 1–5 scale without explicit "No opinion" or "Don't know" options. Consequently, missing data in these columns represents a non-response where the underlying reason (i.e. a lack of opinion or a lack of knowledge on the specific topic) cannot be determined.

## RE-USE POTENTIAL

The dataset has several kinds of potential re-use:

- Characterizing AI practice in bioinformatics: Researchers can quantify which AI tools are used, for which tasks, and with what levels of satisfaction and concern among BioHackathon participants and related communities.
- Comparative studies of AI adoption: the multilingual design and broad field coverage allow comparisons across countries, institution types, research fields, and survey languages. For example, users can examine how translation-related AI use and satisfaction differ between respondents answering in English, Japanese, and Thai.
- AI and workflow design: developers of scientific AI tools, coding assistants, and workflow systems can use the data to identify common pain points, desired improvements, and institutional barriers, and how these vary across linguistic and institutional contexts.
- Methodological and meta-research: the data can support work on AI-in-practice in technical communities, survey design for multilingual settings, and links between institutional policy, language, and AI uptake.

Because both raw anonymized and cleaned English versions are provided, users can choose between higher-fidelity data for qualitative or multilingual work and a compact table for rapid quantitative analysis and cross-tabulation, bearing in mind that the overall sample size is modest.

## DATA AVAILABILITY

All additional supporting data are available in the *GigaScience* repository, GigaDB [2]. The GigaDB record provides access to: (i) raw anonymized data in the original survey languages, (ii) a translated raw English dataset, (iii) a cleaned English analysis table, (iv) a variable-level codebook, (v) a PDF export of the questionnaire, (vi) a translation lookup table. A preprint describing the survey and initial analyses is available via BioHackrXiv/OSF [1], and the GigaDB entry is intended to be cross-linked to this record.

## DECLARATIONS

### Ethics approval and consent to participate

This work is based on an anonymous, voluntary online survey of adults about their professional and related use of AI tools. No names, institutional affiliations, IP addresses, or Google account identities were collected. The questionnaire did not ask about highly sensitive topics such as health status, academic dishonesty or personal misconduct; most items focused on practices, tools, and perceptions in a professional context. Almost all questions were optional, including the "horror stories" item on AI failures, which could be considered more sensitive. An optional email field for follow-up contact was included in the original survey, but no respondents have been contacted and all email addresses were removed prior to dataset release.

Formal ethics committee or institutional review board approval was not sought for this study. The survey introduction informed respondents about the purpose of the survey and that results would be shared back to the community in aggregated form. Completion and submission of the survey were taken as implied consent to participate. Secondary users of the dataset are responsible for ensuring that their use complies with local ethical and governance requirements.

### Consent for publication

Not applicable. The manuscript and accompanying dataset contain only anonymized survey responses and do not include any identifiable individual data or images.

### Competing interests

The authors declare no competing interests.

### Authors' contributions

Conceptualization: DS, CW. Methodology: NS, LF, PT, PN, SG, DS, CW. Investigation: NS, LF, PT, YN, HG, PN, SG, DS, CW. Data curation: NS, LF, CW. Writing - original draft (preprint): NS, LF, PT, YN, HG, PN, SG, DS, CW. Writing - original draft (Data Note): CW. Writing - review & editing: NS, LF, PT, YN, HG, PN, SG, DS, CW. Visualization: NS, LF, PT, YN, HG, SG, PN, CW. Supervision: DS, CW. All authors read and approved the final manuscript.

### Funding

No specific funding was received for the design, data collection, analysis, or writing of this study.

## Acknowledgements

We thank the Database Center for Life Science (DBCLS) for hosting the 2025 BioHackathon. We also thank DBCLS and the Bioinformatics Academic Association of Thailand (BAT) for supporting CW, PT, NS, and PN in attending and contributing to the event. We are grateful to Julia Koblitz (Leibniz Institute DSMZ, Germany) and other 2025 BioHackathon participants for helpful discussions and feedback on the survey.

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
