## [Reviewer Report]

Indicate in the comments box below whether you are happy with the changes made or if the manuscript is unacceptable.Comments on revised manuscriptI've accessed the file and I think the responses they made to the other reviewers are fine. I am ok if you accept at this point.Indicate in the comments box below whether you are happy with the changes made or if the manuscript is unacceptable.Comments on revised manuscriptI've accessed the file and I think the responses they made to the other reviewers are fine. I am ok if you accept at this point.

---

## [Editor Report]

Editor’s AssessmentThe manuscript is ready for formal accept.Editor’s AssessmentThe manuscript is ready for formal accept.

---

## [Reviewer Report]

Reviewer name and names of any other individual's who aided in reviewer Gabrielle O'BrienDo you understand and agree to our policy of having open and named reviews, and having your review included with the published papers. (If no, please inform the editor that you cannot review this manuscript.)YesIs the language of sufficient quality?YesPlease add additional comments on language quality to clarify if needed
Are all data available and do they match the descriptions in the paper? NoAdditional CommentsI don't actually see the dataset provided here, only the PDF of the article. It doesn't appear to be in the OSF repository either (perhaps I am looking in the wrong place? I have never reviewed for GigaByte before so it is possible).Are the data and metadata consistent with relevant minimum information or reporting standards? See GigaDB checklists for examples <a href="http://gigadb.org/site/guide" target="_blank">http://gigadb.org/site/guide</a>NoAdditional CommentsI don't actually see a copy of the data. I would like to see an exact list of the survey questions. It is not sufficient to report the topic they are addressing, the wording is important to have. Please attach a copy of the survey with the exact question wording and wording of response items (for anything that wasn't an open-ended text field).Is the data acquisition clear, complete and methodologically sound?YesAdditional CommentsIs there sufficient detail in the methods and data-processing steps to allow reproduction?NoAdditional CommentsAs commented above, there is insufficient information to reproduce the data collection method because the exact survey questions and response options are not provided (unless I am simply overlooking them?)Is there sufficient data validation and statistical analyses of data quality? NoAdditional CommentsI don't see any statistical analyses of data quality here. It looks like the data has been manually reviewed for quality, which may suffice depending on its size and complexity. But it would be nice to see some counts for completeness.Is the validation suitable for this type of data?YesAdditional CommentsI'm not entirely sure because I don't appear able to see the data itself, but I lean towards yes for the validations that are presented.Is there sufficient information for others to reuse this dataset or integrate it with other data?NoAdditional CommentsAs mentioned above, exact list of survey questions and response options is required and I don't see it.Any Additional Overall Comments to the AuthorRecommendationMajor Revision

---

## [Reviewer Report]

Reviewer name and names of any other individual's who aided in reviewer Pichaya LertvilaiDo you understand and agree to our policy of having open and named reviews, and having your review included with the published papers. (If no, please inform the editor that you cannot review this manuscript.)YesIs the language of sufficient quality?YesPlease add additional comments on language quality to clarify if needed
The manuscript is well-written and the dataset is well-organized for a Data Release publication. The authors have taken appropriate steps toward anonymization and have structured the data for reuse.Are all data available and do they match the descriptions in the paper? YesAdditional CommentsAre the data and metadata consistent with relevant minimum information or reporting standards? See GigaDB checklists for examples <a href="http://gigadb.org/site/guide" target="_blank">http://gigadb.org/site/guide</a>YesAdditional CommentsIs the data acquisition clear, complete and methodologically sound?YesAdditional CommentsSeveral categorical variables in the cleaned dataset contain free-text "Other" responses that fall outside the defined response categories. For example: - **AI usage level:** One response reads `What "AI" are you talking about exactly?! real Artificial-Intelligence or the trendy Artificial-Idiot (TM) !?` and another reads `it is complicated`. These are non-standard responses to what the codebook describes as an "ordinal, single choice" variable. - **Institution type:** Entries include `unemployed` and `Small business owner`, which are outside the defined categories (Academia, Private sector, Public sector). The manuscript does not discuss how these "Other: free-text" responses should be handled in analysis. The codebook mentions that some variables include "Other: free-text" but does not enumerate the free-text values that appear. The authors should either (a) add a note in the codebook or manuscript clarifying how "Other" responses are represented, or (b) consider recoding these in the cleaned dataset.Is there sufficient detail in the methods and data-processing steps to allow reproduction?YesAdditional Comments1. Two column headers in the cleaned data CSV (and both raw data CSVs) contain the typo **"Select all the apply"** instead of **"Select all that apply"**: - Column 2: `What is your field? (Select all the apply)` — codebook has `(Select all that apply)`
- Column 19: `What would AI tools need to improve for you to be more satisfied? (Select all the apply)` — codebook has `(Select all that apply)`
This creates a mismatch between the codebook variable names and the actual CSV headers, which will cause problems for users attempting programmatic joins between the codebook and data files. These typos should be corrected to ensure consistency, or the codebook should match the exact header strings used in the data. 2. The codebook contains 16 entries, but the cleaned dataset has 26 columns. This is because two multi-part questions are represented as single codebook entries: - The "task usage" question expands into 7 sub-columns (Coding, Research, Brainstorming, Writing/Editing, Teaching and curriculum, Translation, Personal use) - The "AI concern" question expands into 5 sub-columns (Bias in algorithms, Data privacy/security, Intellectual property/ownership, Misinformation/Hallucinations, Environment impact) While the codebook mentions the categories, it does not list the 26 individual column names that appear in the CSV. For a dataset intended for reuse, the codebook should ideally have one entry per column (i.e., 26 entries), listing the exact column header as it appears in the CSV file. This would prevent confusion and support automated data dictionary workflows.Is there sufficient data validation and statistical analyses of data quality? YesAdditional Comments1. The cleaned dataset has substantial missing data in several columns that goes undiscussed in the manuscript: - "Is there a reason why you don't use AI?": 99% missing (expected, as this applies only to non-users) - "What would AI tools need to improve?": 68% missing - "Teaching and curriculum" task usage: 41% missing - "Data privacy/security" concern: 39% missing - "Intellectual property/ownership" concern: 33% missing - "Bias in algorithms" concern: 28% - "Environment impact" concern: 28% While the manuscript notes that "almost all questions were optional," the high missing rates for specific items should be mentioned in the Data Validation section, especially if the pattern is non-random (e.g., respondents who skip concern items may differ systematically from those who answer). This information is important for re-users to make informed analytical choices. 2. All three data files (raw original, raw English, cleaned English) contain exactly 105 data rows, confirming structural consistency. However, the manuscript does not report how many total survey starts occurred versus completions, nor whether any responses were excluded (e.g., duplicate submissions, empty submissions). A brief statement on this would strengthen the data validation section. 3. The manuscript states (Data Validation section): *"One response containing an occupation ('Postdoctoral researcher') was set to missing."* This correction was properly applied in the cleaned English file (row 70 is blank), but the raw English file (`Biohackathon2025AISurvey_data_raw_anon_ENG_v1.csv`, row 70) still contains "Postdoctoral researcher" in the country column, as does the original-languages raw file. The manuscript should clarify whether the raw files are intentionally left uncorrected (preserving the original response as-is), or whether this correction was inadvertently omitted from the raw files. If the former, this design decision should be explicitly stated.Is the validation suitable for this type of data?YesAdditional CommentsIs there sufficient information for others to reuse this dataset or integrate it with other data?YesAdditional CommentsThe manuscript acknowledges that "minor spelling and formatting variants were left as entered." However, in the cleaned English dataset (described as "suitable for quantitative analysis"), country values remain unstandardized. For example: - "germany" (lowercase) vs. "Germany" (9 responses) - "U.K." (1 response) vs. "United Kingdom" (2 responses) - "USA" (2 responses) vs. "United States" (1 response) - "Global" (1 response) — an ambiguous non-country value For a dataset intended for quantitative cross-tabulation, these inconsistencies will require downstream users to perform their own harmonization. The authors should consider either standardizing country names in the cleaned dataset or documenting this as a known limitation for re-users.Any Additional Overall Comments to the AuthorThe translation lookup table (Translation_lookup_table_v1.csv) is showing incorrect unicode formatting for Thai and Japanese languages when users download the data set. It is recommended that the authors try to make sure that the format for other languages is well preserved during data upload/download.RecommendationMinor Revision